# RayI2P: Learning Rays for Image-to-Point Cloud Registration

 **Xinjun Li**[1], **Wenfei Yang**[1,*], **Zhixin Cheng**[1], **Jiacheng Deng**[1], **Fei Wang**[3], **Chen Qian**[3], **Tianzhu Zhang**[1,2]
[1]University of Science and Technology of China
[2]National Key Laboratory of Deep Space Exploration, Deep Space Exploration Laboratory
[3]SenseTime Research
{lxj3017,chengzhixin,dengjc,wangfei91}@mail.ustc.edu.cn,
{yangwf,tzzhang}@ustc.edu.cn, qianchen@sensetime.com

## Abstract

Image-to-point cloud registration aims to estimate the 6-DoF camera pose of a query image relative to a 3D point cloud map. Existing methods fall into two categories: matching-free methods regress pose directly using geometric priors, but lack fine-grained supervision and struggle with precise alignment; matching-based methods construct dense 2D-3D correspondences for PnP-based pose estimation, but are fundamentally limited by projection ambiguity (where multiple geometrically distinct 3D points project to the same image patch, leading to ambiguous feature representations) and scale inconsistency (where fixed-size image patches correspond to 3D regions of varying physical size, causing misaligned receptive fields across modalities). To address these issues, we propose a novel ray-based registration framework that first predicts patch-wise 3D ray bundles connecting image patches to the 3D scene and then estimates camera pose via a differentiable ray-guided regression module, bypassing the need for explicit 2D-3D correspondences. This formulation naturally resolves projection ambiguity, provides scale-consistent geometry encoding, and enables fine-grained supervision for accurate pose estimation. Experiments on KITTI and nuScenes show that our approach achieves state-of-the-art registration accuracy, outperforming existing methods.

## 1 Introduction

Image-to-point cloud registration refers to the process of estimating the 6-degree-of-freedom (6-DoF) camera pose of a given 2D image relative to a pre-constructed 3D point cloud map. This task is fundamental to a wide range of computer vision applications, including 3D reconstruction (Dong et al., 2020), AR/VR (Billinghurst et al., 2015), SLAM (Durrant-Whyte & Bailey, 2006), and visual localization (Sarlin et al., 2023). The central challenge lies in the heterogeneous nature of the input modalities: 2D images encode appearance information in dense, regular grids, whereas 3D point clouds represent spatial geometry as sparse, unordered points. This modality gap makes it inherently difficult to design shared feature representations and establish reliable 2D-3D correspondences.

To address this challenge, existing methods can be broadly categorized into matching-free (Li & Lee, 2021; Jeon & Seo, 2022) and matching-based (Ren et al., 2022; Zhou et al., 2023; Wang et al., 2024a; Kang et al., 2024; Bie et al., 2025; Li et al., 2025) approaches. Matching-free methods leverage geometric priors to directly predict camera pose, circumventing the need for explicit correspondence construction. A representative method, DeepI2P (Li & Lee, 2021), formulates image-to-point cloud registration as a combination of frustum classification and pose optimization. It first predicts whether each 3D point lies inside the camera's frustum, and then iteratively adjusts the camera pose to ensure that all predicted in-frustum points are projected within the image plane. However, as illustrated in Figure 1(a), this frustum-based optimization only provides coarse supervision, and the resulting poses are often inaccurate due to the lack of fine-grained alignment. In contrast, matching-based methods have achieved superior performance by establishing dense 2D-3D correspondences, followed by pose estimation using geometric solvers such as PnP-RANSAC (Lepetit et al., 2009;

---

*Corresponding author

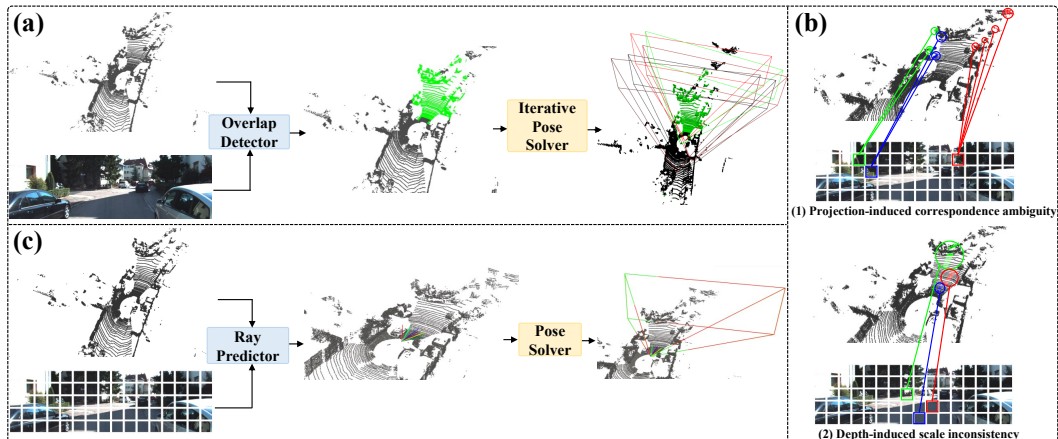

Figure 1: (a) Illustration of the iterative pose optimization process in **matching-free methods** such as DeepI2P. Green points indicate 3D points predicted to lie within the camera frustum. The frustum color transitions from black to red, showing the progression of estimated camera pose during iterative optimization. The green frustum represents ground-truth camera pose. Due to the coarse supervision from frustum classification, the final estimated pose often deviates significantly from the ground truth. (b) Two key challenges of existing **matching-based approaches**: (1) projection-induced correspondence ambiguity: multiple geometrically distinct 3D points project to the same image region; (2) depth-induced scale inconsistency: fixed-size image patches correspond to 3D regions of varying physical size. Each line denotes a ground-truth 2D–3D correspondence between an image patch and a 3D region. (c) Our proposed ray-based registration pipeline predicts a set of 3D rays (colored lines), each representing the projection of an image patch into 3D space. The predicted rays are then used to estimate the camera pose (red frustum), which closely aligns with the ground truth (green frustum). This formulation naturally mitigates the limitations of previous methods.

Fischler & Bolles, 1981). These methods typically extract modality-specific features and align them through attention mechanisms (Vaswani et al., 2017) and contrastive learning (Chopra et al., 2005; Schroff et al., 2015; Sun et al., 2020). Some recent works (Zhou et al., 2023; Wang et al., 2024a) further introduce virtual points or voxels as intermediate representations to bridge the modality gap.

Despite these efforts, matching-based approaches still face two critical challenges: (1) **Projection-induced correspondence ambiguity:** as shown at the top of Figure 1(b), due to perspective projection, a single image pixel or patch may correspond to multiple spatially disjoint 3D points distributed along a viewing ray. These 3D points can differ significantly in geometry (e.g., curvature, surface normals, or semantics), yet they are all mapped to the same image region. As a result, models are forced to align geometrically dissimilar 3D features to a single image feature, leading to ambiguous or collapsed feature representations. This undermines the learning of discriminative similarity metrics and results in unreliable 2D-3D correspondences, ultimately harming pose estimation accuracy. (2) **Depth-induced scale inconsistency**: as shown at the bottom of Figure 1(b), fixed-size image patches can correspond to 3D regions of vastly different physical scales depending on their depth. For instance, a small object nearby and a large object in the distance may occupy similar image areas but represent vastly different 3D scales. This causes a mismatch in receptive field alignment: image features are extracted from fixed local neighborhoods, while corresponding 3D features vary significantly in spatial extent depending on depth. Such scale inconsistency makes it difficult to learn scale-consistent similarity metrics and establish semantically meaningful matches. Together, these issues limit the reliability and generalization capability of learned correspondences, especially in complex outdoor environments.

**In this work, we take a fundamentally different perspective: instead of modeling 2D-3D correspondences directly, we model *rays* that implicitly connect image patches to the 3D scene.** We observe that under the pinhole camera model, each image patch naturally corresponds to a potential ray in 3D space, originating from the camera center and extending into the 3D scene. By learning to predict these rays, we can effectively bridge the modality gap and enable accurate pose estimation without relying on explicit 2D-3D matching. To realize this idea, we propose a novel ray-based framework for image-to-point cloud registration as shown in Figure 1(c). Specifically, we design a two-stage ray-based cross-modal registration pipeline: a ray prediction module that integrates image

and point cloud features via attention mechanisms (Vaswani et al., 2017) to infer consistent 3D rays for each image patch, and a differentiable ray-guided pose regression module that jointly estimates camera's rotation and translation from predicted ray bundles. Our ray-based representation provides several key advantages. First, it resolves projection-induced correspondence ambiguity by replacing discrete pixel-to-point matching with a continuous ray formulation, where each image patch is interpreted as a 3D ray that encodes both its viewing direction and origin in the point cloud. Second, it alleviates depth-induced scale inconsistency by modeling scene geometry in terms of directional cues rather than fixed spatial extents. Since ray directions are invariant to the depth of the underlying structure, our approach maintains consistent representations regardless of whether a patch corresponds to a nearby object or a distant surface. Third, associating each image patch with a ray bundle provides fine-grained geometric supervision. Rather than merely predicting whether a 3D point is visible, our model learns how each image patch projects into space, yielding stronger and more spatially aware signals during training. This finer-grained modeling enhances pose accuracy, especially in scenes with complex geometry or partial observations. Despite the overall simplicity, extensive experiments on KITTI and nuScenes show that our framework achieves state-of-the-art registration performance, consistently outperforming existing matching-free and matching-based methods.

The main contributions are summarized as follows: (1) We propose a novel ray-based paradigm for image-to-point cloud registration, which effectively addresses the core limitations of prior approaches by modeling image patches as continuous 3D ray bundles, thereby resolving projection-induced correspondence ambiguity and depth-induced scale inconsistency, and enabling fine-grained, direction-aware pose supervision beyond coarse geometric constraints. (2) Extensive experiments on KITTI and nuScenes demonstrate that our method achieves state-of-the-art performance in cross-modal registration accuracy, validating the effectiveness of our ray-based representation.

## 2 RELATED WORKS

In this section, we provide a brief overview of related works on image-to-image registration, point cloud-to-point cloud registration, and image-to-point cloud registration.

**Image-to-Image Registration**. This task aims to align images of the same scene taken under varying conditions (e.g., time, viewpoint) by estimating a spatial transformation. Existing methods fall into two main categories: detection-based and detection-free. Detection-based methods extract and match sparse keypoints, either hand-crafted (Lowe, 2004; Rublee et al., 2011) or learned (Gao et al., 2023; DeTone et al., 2018; Sarlin et al., 2020; Dusmanu et al., 2019; Revaud et al., 2019), but often struggle in low-texture regions. Detection-free methods (Shen et al., 2023; Zhou et al., 2021; Sun et al., 2021; Wang et al., 2024b) bypass keypoint detection by estimating coarse-to-fine patch- and pixel-level correspondences, enabling robust alignment.

**Point Cloud-to-Point Cloud Registration**. The goal here is to estimate a rigid transformation between two 3D point clouds. Traditional approaches rely on matching sparse keypoints (Drost et al., 2010; Rusu et al., 2009; Bai et al., 2020; Deng et al., 2018), but are sensitive to density variations, occlusions, and noise. Detector-free methods (Yu et al., 2021; Lu et al., 2023; Qin et al., 2023) instead learn dense or semi-dense correspondences directly from global geometric context. CoFiNet (Yu et al., 2021), for example, uses a coarse-to-fine pipeline without explicit keypoint detection. Recent works further integrate learned pose estimators (Qin et al., 2023) to replace classical RANSAC (Fischler & Bolles, 1981), achieving better robustness.

**Image-to-Point Cloud Registration**. Compared to intra-modal registration, this task is more challenging due to the modality gap. A common pipeline involves establishing 2D-3D correspondences followed by pose estimation using PnP-RANSAC (Lepetit et al., 2009; Fischler & Bolles, 1981). Early methods such as 2D3D-MatchNet (Feng et al., 2019) first use modality-specific keypoint detectors (Lowe, 2004; Zhong, 2009) to get keypoints and then extract keypoint features via CNNs (Simonyan & Zisserman, 2015) or PointNet (Qi et al., 2017). These independent hand-crafted keypoint detectors for different modalities leads to a poor registration accuracy. Recent learning-based approaches propose to predict dense 2D-3D correspondences. CorrI2P (Ren et al., 2022) estimates overlapping regions and matches 2D/3D features, while VP2P-Match (Zhou et al., 2023) adopts voxel-based representations and differentiable PnP (Chen et al., 2022), though both are still sensitive to modality discrepancies. CoFiI2P (Kang et al., 2024) follows the design of LoFTR (Sun et al., 2021) but overlooks modality gaps, limiting its generalization. ICL (Li et al., 2025) intro-

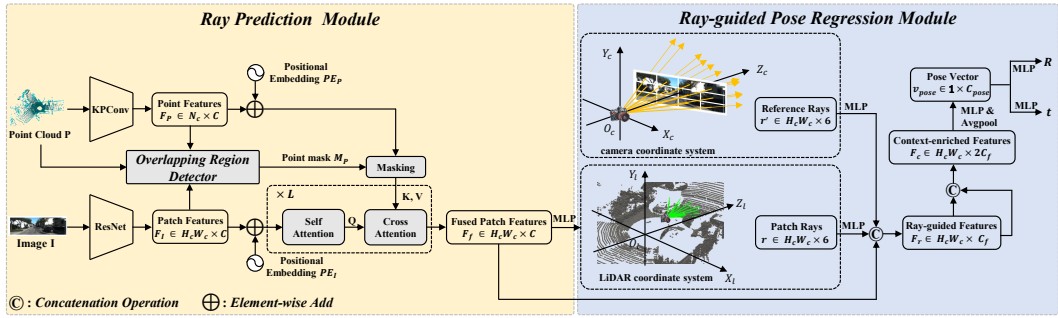

Figure 2: Overview of our proposed image-to-point cloud registration framework. Given a pair of image $\mathbf{I}$ and point cloud $\mathbf{P}$, we first extract downsampled patch features $\mathbf{F}_I$ and point features $\mathbf{F}_P$, respectively. An overlapping region detector then predicts a binary mask $\mathbf{M}_P$ indicating whether each 3D point lies within the camera frustum. Next, cross-modal attention fuses patch and point features into enriched patch features $\mathbf{F}_f$. Based on $\mathbf{F}_f$, a lightweight MLP predicts 3D rays $\mathbf{r}$ for each image patch, representing potential projection in 3D space. Finally, the ray-guided pose regression module estimates camera pose $(\mathbf{R}, \mathbf{t})$ by jointly leveraging predicted patch rays $\mathbf{r}$, reference rays $\mathbf{r}'$ (computed from camera intrinsic), and fused patch features $\mathbf{F}_f$.

duces an implicit correspondence learning framework for direct pose regression, yet its reliance on weak geometric priors hampers performance under large viewpoint changes or sparse inputs. GraphI2P (Bie et al., 2025) incorporates external depth prediction (Bhat et al., 2023) to reduce domain gaps, but incurs high computational cost. Beyond matching-based methods, some approaches directly predict camera pose without explicit correspondence construction by leveraging geometric priors. DeepI2P (Li & Lee, 2021), for instance, reformulates the task as a frustum classification followed by inverse projection. However, its coarse 2D-3D associations often fail to achieve high-precision alignment.

## 3 METHOD

### 3.1 OVERVIEW

Given an image $\mathbf{I} \in \mathbb{R}^{H \times W \times 3}$ and a point cloud $\mathbf{P} \in \mathbb{R}^{N \times 3}$ from the same scene, our goal is to determine the camera pose $\mathbf{T}_{gt}$ in coordinate system of $\mathbf{P}$, which consists of a rotation matrix $\mathbf{R}_{gt} \in \mathbf{SO(3)}$ and a translation vector $\mathbf{t}_{gt} \in \mathbb{R}^3$. In this paper, we propose a ray-based image-to-point cloud registration method composed of two main stages: a ray prediction module to infer consistent 3D rays for each image patch, and a differentiable ray-guided pose regression module that jointly estimates camera's rotation and translation from predicted ray bundles, as shown in Figure 2.

### 3.2 BACKGROUND: RAY-BASED CAMERA REPRESENTATION

Traditional cameras are usually parameterized by extrinsic rotation $\mathbf{R} \in \mathbf{SO(3)}$, translation $\mathbf{t} \in \mathbb{R}^3$, and intrinsic calibration $\mathbf{K} \in \mathbb{R}^{3 \times 3}$. While compact, this low-dimensional representation is difficult to regress directly from complex visual and geometric features because of strong geometric constraints and nonlinearities. To address this difficulty, we instead adopt a more expressive formulation by representing the camera as a bundle of rays associated with image patches, inspired by generalized camera models (Grossberg & Nayar, 2001; Schops et al., 2020) and recent ray-based representations (Zhang et al., 2024), Each image patch, centered at pixel coordinate $\mathbf{u}_i$, corresponds to a 3D ray $\mathbf{r}_i \in \mathbb{R}^6$ encoded in Plücker coordinates (Plücker, 1828):

$$\mathbf{r}_i = [\mathbf{d}_i, \mathbf{m}_i], \quad \text{where} \quad \mathbf{m}_i = \mathbf{p}_i \times \mathbf{d}_i. \tag{1}$$

Here, $\mathbf{d}_i \in \mathbb{R}^3$ denotes the ray's direction, $\mathbf{p}_i \in \mathbb{R}^3$ is any 3D point on this ray, and $\mathbf{m}_i \in \mathbb{R}^3$ is the associated moment vector, invariant to the choice of $\mathbf{p}_i$. Notably, when $\mathbf{d}_i$ is normalized, the norm of the moment $\mathbf{m}_i$ represents the distance of the ray from the origin.

Given camera parameters $(\mathbf{R}, \mathbf{t}, \mathbf{K})$, we can generate rays by unprojecting image patches into 3D:

$$\mathbf{d}_i = \mathbf{R}^\top \mathbf{K}^{-1} \tilde{\mathbf{u}}_i, \quad \mathbf{m}_i = (-\mathbf{R}^\top \mathbf{t}) \times \mathbf{d}_i, \tag{2}$$

where $\tilde{\mathbf{u}}_i \in \mathbb{R}^3$ is the homogeneous coordinate of $\mathbf{u}_i$. The direction $\mathbf{d}_i$ points from the camera center into the 3D scene, and the moment $\mathbf{m}_i$ captures the ray's offset from the origin.

Conversely, a camera can be approximately recovered from predicted rays $\mathbf{r}_i = [\mathbf{d}_i, \mathbf{m}_i]$. Camera center $\mathbf{c}$ is obtained by minimizing distances from a point to all rays:

$$\mathbf{c} = \arg\min_{\mathbf{p} \in \mathbb{R}^3} \sum_i \|\mathbf{p} \times \mathbf{d}_i - \mathbf{m}_i\|^2. \tag{3}$$

The rotation $\mathbf{R}$ is then estimated by aligning ray directions with pixel vectors:

$$\mathbf{P} = \arg\min_{\|\mathbf{H}\|=1} \sum_i |\mathbf{H}\mathbf{d}_i \times \tilde{\mathbf{u}}_i|, \tag{4}$$

where the homography matrix $\mathbf{P}$ can be decomposed into $\mathbf{K}$ and $\mathbf{R}$ via RQ-decomposition. Finally, the translation vector is recovered as $\mathbf{t} = -\mathbf{R}^\top \mathbf{c}$. By bridging compact camera parameters and over-parameterized ray bundles, this two-way conversion yields a representation that combines geometric interpretability with modeling flexibility, serving as the cornerstone of our framework.

### 3.3 Ray Prediction Module

**Feature Extraction**. For the input image $\mathbf{I} \in \mathbb{R}^{H \times W \times 3}$, we use ResNet (He et al., 2016) to extract downsampled 2D features and flatten them to obtain patch features $\mathbf{F}_I \in \mathbb{R}^{H_c W_c \times C}$, where $H_c = \frac{1}{8}H, W_c = \frac{1}{8}W$. For the input point cloud $\mathbf{P} \in \mathbb{R}^{N \times 3}$, we use KPConv (Thomas et al., 2019) to extract downsampled point features $\mathbf{F}_P \in \mathbb{R}^{N_c \times C}$. We also record the corresponding downsampled patch coordinates $\mathbf{E}_I \in \mathbb{R}^{H_c W_c \times 2}$ and point coordinates $\mathbf{E}_P \in \mathbb{R}^{N_c \times 3}$.

**Overlapping Region Detector**. To avoid ambiguous supervision from non-overlapping 3D points, we adopt the strategy from ICL (Li et al., 2025), with a minor modification of adding 3D coordinates as input. The resulting binary mask $\mathbf{M}_P$ indicates visible 3D points within the camera view.

**Ray Prediction via Cross-Modal Feature Fusion**. In this part, our goal is to predict associated 3D rays for each image patch by fusing patch features with relevant 3D information from the point cloud. Given extracted patch features $\mathbf{F}_I$ and point features $\mathbf{F}_P$, we first introduce learnable positional embeddings to encode spatial cues. Specifically, we map patch coordinates $\mathbf{E}_I$ into patch positional embedding $\mathbf{PE}_I$ using a MLP. Likewise, point coordinates $\mathbf{E}_P$ are mapped into point positional embedding $\mathbf{PE}_P$. The resulting embeddings are added to the corresponding patch and point features to produce position-aware representations:

$$\mathbf{F}'_I = \mathbf{F}_I + \mathbf{PE}_I, \quad \mathbf{F}'_P = \mathbf{F}_P + \mathbf{PE}_P. \tag{5}$$

To suppress interference from irrelevant geometry, we apply predicted point mask $\mathbf{M}_P$ to retain only visible points from the point cloud. We then apply a transformer-based fusion module (Vaswani et al., 2017) consisting of multiple self and cross attention layers, executed in an alternate fashion for $L$ iterations. In each iteration, a self attention layer allows image patches to exchange contextual information within the image, followed by a cross attention layer where each patch attends to the visible 3D points. Through $L$ rounds of alternate interaction, the patch features are progressively refined with both global image context and geometry-aware cues from the point cloud, enabling the network to reason about each patch's potential ray in 3D space. Finally, the fused patch features $\mathbf{F}_f$ are passed through a MLP head to predict patch rays $r \in \mathbb{R}^{H_c W_c \times 6}$ for every patch, representing its origin and direction in 3D space.

**Focus loss**. To encourage each image patch to attend more to geometrically relevant 3D points, we propose a focus loss that guides the attention distribution in cross attention layers. Specifically, we encourage each image patch to assign higher attention scores to 3D points whose projections fall within a local neighborhood (a circle with radius of $\sigma$ pixels) centered at the patch. Let $\mathbf{H} \in \mathbb{R}^{H_c W_c \times N_c}$ denotes the averaged attention map across all cross-attention layers, $\mathbf{E}_{I,i} \in \mathbb{R}^2$ be the center coordinate of image patch $i$, and $\mathbf{E}_{P,j}^{2D} \in \mathbb{R}^2$ be the 2D projection of point $j$ in the image plane. We define an indicator function:

$$\mathbf{1}_{ij} = \begin{cases} 1, & \text{if } \|\mathbf{E}_{I,i} - \mathbf{E}_{P,j}^{2D}\|_2 < \sigma \\ 0, & \text{otherwise} \end{cases}, \tag{6}$$

Then the focus loss is formulated as:

$$\mathcal{L}_{foc} = 1 - \frac{1}{H_c W_c} \sum_{i=1}^{H_c W_c} \sum_{j=1}^{N_c} \mathbf{H}_{ij} \cdot \mathbf{1}_{ij}. \qquad (7)$$

This loss guides attention toward spatially local, geometrically meaningful regions, enhancing ray prediction accuracy and accelerating convergence.

## 3.4 RAY-GUIDED POSE REGRESSION MODULE

In practice, predicted rays may suffer from noise, ambiguity, or misalignment. As a result, directly applying a geometric solver, as described in Equation 3 and 4, may lead to unstable or inaccurate pose estimations. We provide visual evidence of this instability in the Appendix 5. To address this, we propose a learnable ray-guided pose regression module that estimates the camera pose from fused patch features $\mathbf{F}_f$, predicted patch rays $\mathbf{r}$, and reference rays $\mathbf{r}'$ in a ray-guided and fully differentiable manner. The predicted patch rays $r$ are defined in LiDAR coordinate system, whereas the reference rays $\mathbf{r}' \in \mathbb{R}^{H_c W_c \times 6}$ represent patch-associated rays in camera coordinate system. The reference rays are computed from known camera intrinsic $\mathbf{K}$ and patch coordinates $\mathbf{E}_I$, applying Equation 2 with the rotation matrix $\mathbf{R} = \mathbf{I}$ and translation vector $\mathbf{t} = \mathbf{0}$. These rays serve as geometric anchors to guide pose regression. Our ablation study shows that this learnable formulation outperforms geometric solvers in both accuracy and robustness.

Specifically, we first apply MLPs to transform fused patch features $\mathbf{F}_f$, predicted patch rays $\mathbf{r}$ and reference rays $\mathbf{r}'$, and concatenate them to obtain the ray-guided features $\mathbf{F}_r$. This fusion provides the network with a unified representation that combines cues from fused patch features and directional constraints from both predicted and reference rays, enabling reliable and spatially grounded pose estimation. To incorporate global spatial context, we perform average pooling over $\mathbf{F}_r$ and concatenate the pooled result back to $\mathbf{F}_r$, forming context-enriched features $\mathbf{F}_c$. These are further refined by a MLP and pooled again to yield a compact pose representation vector $\mathbf{v}_{pose}$. Finally, two lightweight MLP heads take $\mathbf{v}_{pose}$ as input and predict the rotation $\mathbf{R}$ and translation $\mathbf{t}$, respectively. Here, we adopt the 6D representation of rotation (Zhou et al., 2019) to parameterize rotation due to its continuity and suitability for learning in $\mathbf{SO(3)}$.

## 3.5 LOSS FUNCTION

Given ground-truth camera pose $(\mathbf{R}_{gt}, \mathbf{t}_{gt})$ and intrinsic matrix $\mathbf{K}$, we design a composite loss to jointly supervise the model. The overall loss consists of three terms: a ray regression loss $\mathcal{L}_{ray}$, a camera pose loss $\mathcal{L}_{cam}$, and a focus loss $\mathcal{L}_{foc}$ introduced in Equation 7. Each term is tailored to supervise a specific sub-task, collectively guiding the model towards accurate and robust registration.

**Ray regression loss**. To supervise ray prediction, we design a ray-level regression loss tailored to our setting. Given ground-truth camera parameters, ground truth ray bundles $r_{gt}$ can be computed via Equation 2. We apply an $L_2$ loss over all patches to enforce accurate ray alignment:

$$\mathcal{L}_{ray} = \frac{1}{H_c W_c} \sum_{i=1}^{H_c W_c} \|r_{gt,i} - r_i\|_2. \qquad (8)$$

**Camera pose loss**. We define camera pose loss $\mathcal{L}_{cam}$ to directly supervise predicted camera pose:

$$\mathcal{L}_{cam} = \|\mathbf{R}_{gt} - \mathbf{R}\|_2 + \|\mathbf{t}_{gt} - \mathbf{t}\|_2. \qquad (9)$$

To jointly optimize all components, we define the total loss as the sum of the three sub-losses:

$$\mathcal{L}_{total} = \mathcal{L}_{ray} + \mathcal{L}_{cam} + \mathcal{L}_{foc}. \qquad (10)$$

## 4 EXPERIMENTS

### 4.1 IMPLEMENTATION DETAILS

In this work, we implement the proposed model in Pytorch (Paszke et al., 2019) and adopt a single NVIDIA RTX 3090 GPU for training. We adopt a 4-stage ResNet (He et al., 2016) as the image

Table 1: Registration accuracy on the KITTI and nuScenes datasets. Here † represents method that adopts external powerful depth estimation model (Bhat et al., 2023) for image depth estimation.

| Category | Method | KITTI | | | nuScenes | | |
|---|---|---|---|---|---|---|---|
| | | RTE(m)↓ | RRE(°)↓ | Acc(%)↑ | RTE(m)↓ | RRE(°)↓ | Acc(%)↑ |
| Matching-based | CorrI2P (Ren et al., 2022) | 3.78±65.16 | 5.89±20.34 | 72.42 | 3.04±60.76 | 3.73±9.03 | 49.00 |
| | VP2P-match (Zhou et al., 2023) | 0.75±1.13 | 3.29±7.99 | 83.04 | 0.89±1.44 | 2.15±7.03 | 88.33 |
| | FreeReg† (Wang et al., 2024a) | 0.95±1.05 | 2.06±3.21 | 91.68 | - | - | - |
| | CoFiI2P (Kang et al., 2024) | 0.31±0.20 | 1.24±0.84 | - | 1.21±9.55 | 2.54±8.93 | - |
| | ICL (Li et al., 2025) | 0.20±0.21 | 1.24±2.34 | 97.49 | 0.63±0.44 | 2.13±3.75 | 90.94 |
| | GraphI2P† (Bie et al., 2025) | 0.32±0.81 | 1.65±1.32 | 99.61 | 0.49±1.22 | 1.73±**1.63** | **99.48** |
| Matching-free | Grid Cls. + PnP (Li & Lee, 2021) | 3.64±3.46 | 19.19±28.96 | 11.22 | 3.02±2.40 | 12.66±21.01 | 2.45 |
| | DeepI2P(3D) (Li & Lee, 2021) | 4.06±3.54 | 24.73±31.69 | 3.77 | 2.88±2.12 | 20.65±12.24 | 2.26 |
| | DeepI2P(2D) (Li & Lee, 2021) | 3.59±3.21 | 11.66±18.16 | 25.95 | 2.78±1.99 | 4.80±6.21 | 38.10 |
| | Ours | **0.09±0.08** | **0.63±0.71** | **99.75** | **0.39±0.29** | **1.48**±5.72 | 96.61 |

| Input | VP2P-match | ICL | Ours | GT |
|---|---|---|---|---|

| 10.63m / 83.36° | 2.47m / 22.84° | 1.24m / 23.67° | 0.09m / 0.35° | RTE(m) / RRE(°) |
| 5.16m / 51.40° | 3.31m / 19.58° | 1.17m / 26.42° | 0.11m / 0.31° | RTE(m) / RRE(°) |
| 9.11m / 64.44° | 1.25m / 28.99° | 1.51m / 39.09° | 0.11m / 0.67° | RTE(m) / RRE(°) |

Figure 3: Qualitative comparison of Image-to-Point Cloud registration results on KITTI dataset.

backbone network, where the output channel dimension is 512. For the 3D backbone, we use a 4-stage KPConv (Thomas et al., 2019) where the output channel dimension is also 512. We set the channel dimension $C_f$ to 256 and set $C_{pose}$ to 512. All the attention layers have 512 features channels with 4 attention heads and ReLU activation. The default iteration time $L$ and focus radius $\sigma$ are set to 2 and 32, respectively. All parameters in the proposed model are randomly initialized and trained from scratch with the Adam optimizer. We train the whole network with the total loss $\mathcal{L}_{total}$ for 20 epochs. The learning rate is set to $10^{-4}$, and the weight decay is set to be $10^{-6}$.

## 4.2 DATASETS

We conduct experiments on two mostly used benchmarks: KITTI and nuScenes.

**KITTI Odometry** (Geiger et al., 2013). KITTI Odometry dataset comprises 22 driving sequences collected in urban scenarios, with 11 of them providing ground-truth calibration files. Following standard protocol (Li & Lee, 2021), we use sequences 0–8 for training and sequences 9–10 for testing. To simulate misalignment, we apply random 2D translations within ±10m on the ground plane and arbitrary yaw-axis rotations. All input images are resized to a resolution of $160 \times 512$, and point clouds are uniformly downsampled to 40,960 points for both training and evaluation.

**nuScenes** (Caesar et al., 2020). We generate image-point cloud pairs using the official nuScenes SDK, where point clouds are aggregated from nearby LiDAR frames, while images are taken from the current frame. We follow the official train/test split, using 850 scenes for training and 150 scenes for testing. The same mis-registration strategy as in KITTI is applied. For consistency, we downsample the image resolution to $160 \times 320$ and retain 40,960 points per point cloud.

## 4.3 EVALUATION METRICS

To assess registration performance, we follow the protocol from VP2P-match (Zhou et al., 2023), reporting three key metrics: average Relative Translation Error (RTE), average Relative Rotation Error (RRE), and registration accuracy (Acc). Unlike CorrI2P (Ren et al., 2022), which filters out high-error samples before computing averages, we retain all test pairs during evaluation to better reflect real-world robustness. Following VP2P-match (Zhou et al., 2023), we define Acc as the proportion of samples where the estimated transformation achieves RTE < 2m and RRE < 5°.

Predicted Camera              Attention map

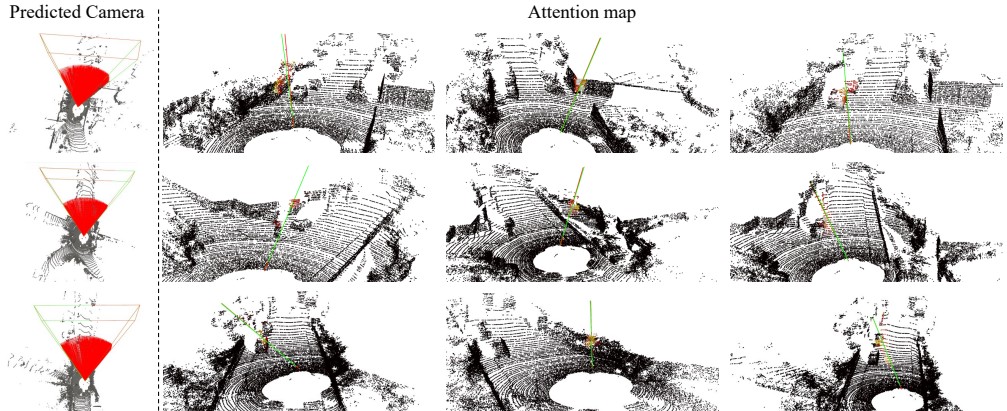

Figure 4: **Visualization of predicted and ground truth camera poses, rays, and attention maps.** Left: Predicted camera poses (red) and GT poses (green), with red lines representing predicted rays for all image patches. Right: Predicted rays (red) and GT rays (green) overlaid on 3D point cloud, along with attention maps highlighting which 3D regions this image patch attends to. Our model produces accurate ray predictions and pose estimations, while attending to geometrically meaningful regions, showcasing the effectiveness and interpretability of our ray-guided registration framework.

## 4.4 COMPARISON WITH STATE-OF-THE-ART METHODS

**Baselines**. Table 1 lists several representative matching-free and matching-based baselines for comparison with our method.

**Quantitative Comparison.** Table 1 summarizes the quantitative performance of our method against existing approaches. On the KITTI dataset, our method consistently outperforms all baselines across all evaluation metrics, including GraphI2P, which benefits from an auxiliary high-quality depth estimator. Specifically, our method achieves the highest registration accuracy (Acc), while reducing RTE by 0.11m and RRE by 0.61° compared to ICL. On the nuScenes dataset, our approach also shows strong generalization, surpassing ICL by 0.24m in RTE and 0.65° in RRE. These results highlight the effectiveness of our ray-based representation and demonstrate that a simple architecture, combined with our formulation, is sufficient to achieve accurate image-to-point cloud registration.

**Efficiency comparison of different methods.** In Table 2, we provide a comparison of model size and inference time across different methods, with results obtained on the same machine using a single RTX 3090 GPU. Methods such as DeepI2P (Li & Lee, 2021) and CorrI2P (Ren et al., 2022), which include time-consuming post-

Table 2: Efficiency comparisons.

| Method | Model size (MB) | Inference Time (s) |
|---|---|---|
| DeepI2P(2D) (Li & Lee, 2021) | 100.12 | 23.47 |
| DeepI2P(3D) (Li & Lee, 2021) | 100.12 | 35.61 |
| CorrI2P (Ren et al., 2022) | 141.07 | 8.96 |
| VP2P-match (Zhou et al., 2023) | **30.73** | 0.19 |
| CoFiI2P (Kang et al., 2024) | 39.09 | 0.18 |
| ICL (Li et al., 2025) | 45.21 | 0.24 |
| Ours | 49.24 | **0.11** |

processing steps for pose estimation, show significantly slower inference speeds. In contrast, our method achieves an inference speed approximately 80× (or more) faster, showcasing a clear efficiency advantage. VP2P-match (Zhou et al., 2023), CoFiI2P (Kang et al., 2024), and ICL (Li et al., 2025) all perform computations at the original resolution or in a coarse-to-fine manner, which increases their computational cost and inference time. In contrast, our method operates at a downsampled resolution, significantly reducing the computational demand while maintaining similar model parameters. As a result, our method achieves much faster inference time, making it more efficient without compromising performance.

**Qualitative Comparison.** Figure 3 presents qualitative comparisons between our method and representative baselines. To facilitate visual inspection, we project the point cloud onto image plane using estimated transformation and known camera intrinsic, with point color encoding actual depth value. Across diverse road scenes, our method achieves superior registration accuracy compared to VP2P-match (Zhou et al., 2023) and ICL (Li et al., 2025), demonstrating the effectiveness of

Table 3: Ablation studies on Ray-guided Pose Regression Module. FPF: Fused Patch Features $\mathbf{F}_f$; PR: Patch Rays $\mathbf{r}$; RR: Renference Rays $\mathbf{r}'$; CPS: classical pose solver (Equation 3 and 4). ✓: included, ✗: excluded.

| FPF | PR | RR | CPS | KITTI | | | NuScenes | | |
|---|---|---|---|---|---|---|---|---|---|
| | | | | RTE(m)↓ | RRE(°)↓ | Acc(%)↑ | RTE(m)↓ | RRE(°)↓ | Acc(%)↑ |
| ✓ | ✗ | ✗ | ✗ | 0.33±0.18 | 2.07±1.67 | 94.48 | 0.48±0.32 | 3.23±10.05 | 89.79 |
| ✗ | ✓ | ✗ | ✗ | 0.34±0.18 | 1.14±1.16 | 98.66 | 0.50±0.32 | 2.27±8.88 | 93.75 |
| ✓ | ✓ | ✗ | ✗ | 0.10±0.09 | 0.73±0.91 | 99.41 | 0.41±0.29 | 2.10±8.44 | 94.25 |
| ✗ | ✓ | ✓ | ✓ | 0.10±**0.08** | 0.82±0.78 | 99.62 | **0.39**±**0.28** | 2.51±20.93 | 94.49 |
| ✗ | ✓ | ✓ | ✗ | 0.10±**0.08** | 0.74±0.75 | 99.64 | 0.43±0.33 | 1.89±6.32 | 95.38 |
| ✓ | ✓ | ✓ | ✗ | **0.09**±**0.08** | **0.63**±**0.71** | **99.75** | **0.39**±0.29 | **1.48**±**5.72** | **96.61** |

our ray-based formulation in enabling more reliable and structure-aware registration. We further visualize predicted camera poses and corresponding predicted rays alongside ground-truth values. As shown in Figure 4, our method produces rays that are closely aligned with the ground truth, and estimated camera poses exhibit minimal deviation from the ground truth. To better understand the cross-modal interaction, we also visualize attention maps for individual rays. These visualizations confirm that the model learns to attend to geometrically consistent regions in the point cloud, validating the interpretability and effectiveness of our attention-guided ray prediction mechanism.

### 4.5 Ablation Study

**Ablation on Ray-guided Pose Regression Module.** To better understand the contribution of each component in our Ray-guided Pose Regression Module, we conduct ablation studies by selectively removing or replacing fused patch features (FPF), patch rays (PR), reference rays (RR), and classical pose solver (CPS). As shown in Table 3, using only fused patch features (Row 1) performs poorly, while introducing patch rays (Row 2) brings a large improvement, highlighting the benefit of our ray-based camera representation. Combining fused patch features with patch rays (Row 3) yields further gains, showing their complementarity. While classical pose solver (Row 4) achieves reasonable results, it is less stable than learning-based formulation, with notably larger mean and variance in rotation errors on the nuScenes dataset. Additional visual comparisons are provided in the Appendix 5, further illustrating the instability of directly applying a classical pose solver. In contrast, incorporating both patch rays and reference rays into the learnable regression framework (Row 5), the model shows improved robustness. The full model (Row 6) further achieves the best overall performance, highlighting both the necessity of a learnable regression module and the contribution of each component in our Ray-guided Pose Regression Module.

**Ablation on Focus Loss $\mathcal{L}_{foc}$.** We investigate the effect of focus radius $\sigma$, which governs the spatial constraints in cross-attention between patch and point features. As shown in Table 4, increasing $\sigma$ from 8 to 32 enhances performance, indicating that a larger local neighborhood provides richer 3D context for registration. However, further increasing $\sigma$ beyond 32, along with the removal of the focus loss, leads to performance degradation. This is likely due to the inclusion of irrelevant or noisy points, which diminish the effectiveness of focused interactions. Overall, $\sigma = 32$ strikes the best balance between geometric context and locality, and is therefore used in all experiments. More detailed ablation results are provided in Appendix 6.

Table 4: Ablation studies on focus radius $\sigma$. Here ✗ indicates method that doesn't use $\mathcal{L}_{foc}$.

| $\sigma$ | KITTI | | | nuScenes | | |
|---|---|---|---|---|---|---|
| | RTE(m)↓ | RRE(°)↓ | Acc(%)↑ | RTE(m)↓ | RRE(°)↓ | Acc(%)↑ |
| ✗ | 0.10±0.09 | 1.02±1.00 | 99.18 | 0.42±0.31 | 1.94±6.22 | 94.44 |
| 8 | 0.11±0.09 | 0.75±0.95 | 99.30 | 0.41±0.30 | 1.87±6.39 | 94.70 |
| 32 | **0.09**±**0.08** | **0.63**±**0.71** | **99.75** | **0.39**±**0.29** | **1.48**±**5.72** | **96.61** |
| 128 | **0.09**±0.09 | 0.91±0.86 | 99.43 | 0.41±**0.29** | 1.77±6.57 | 95.43 |

## 5 Conclusion

In this paper, we present a novel ray-based framework for image-to-point cloud registration that overcomes key limitations of both matching-based and matching-free approaches. By modeling

each image patch as a 3D ray, our method learns a ray-guided representation that captures continuous and spatially consistent geometric relationships across modalities. The framework employs a two-stage pipeline: cross-modal attention to predict dense ray bundles, followed by differentiable ray-guided pose regression. This design effectively mitigates projection ambiguity and scale inconsistency, while offering stronger geometric cues for accurate pose estimation. Extensive experiments on KITTI and nuScenes show that our simple yet effective architecture achieves state-of-the-art performance in both accuracy and robustness. These results highlight the potential of ray-based reasoning for bridging the gap between images and point clouds in cross-modal registration.

## ACKNOWLEDGEMENTS

This work was partially supported by the National Natural Science Foundation of China (NO. 6230070181,42595545) and Youth Innovation Promotion Association of CAS.

## ETHICS STATEMENT

This work adheres to the ICLR Code of Ethics. Our research does not involve human subjects, sensitive personal data, or practices that may raise immediate ethical concerns. All datasets used are publicly available and widely adopted in the community, and we ensure that the use of these datasets complies with their respective licenses. The proposed methodology is intended for advancing research in vision and robotics, and we do not foresee direct misuse or harmful applications beyond the standard risks associated with general-purpose machine learning techniques. In addition, our study avoids introducing biases related to gender, race, or socioeconomic factors, and we maintain transparency by documenting the experimental setup and implementation details. We are committed to open research practices and will release all code, datasets, and model weights to the community to support ethical and responsible use of our contributions.

## REPRODUCIBILITY STATEMENT

We have made extensive efforts to ensure the reproducibility of our work. Detailed descriptions of the model architecture, training strategies, and hyperparameters are provided in the main text and appendix. All derivations and proofs are included for theoretical results where appropriate. The datasets used in our experiments are publicly accessible, and we include evaluation protocols in the appendix. To further facilitate reproducibility, we will release the complete implementation, trained model weights, and instructions for data preparation upon acceptance. Together, these resources will allow the community to reproduce our results and extend our work with minimal effort.

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

# A APPENDIX

## A.1 EVALUATION METRICS

To quantitatively evaluate the performance of image-to-point cloud registration, we adopt two widely used metrics that measure the accuracy of estimated 6-DoF (Degrees of Freedom) transformation: **Relative Translation Error (RTE)** and **Relative Rotation Error (RRE)**. RTE assesses the deviation in translation, while RRE quantifies the discrepancy in rotation between predicted and ground-truth camera poses.

### A.1.1 RELATIVE TRANSLATION ERROR (RTE)

RTE measures the Euclidean distance between predicted translation vector $\mathbf{t}$ and ground-truth translation vector $\mathbf{t}_{gt}$, thereby reflecting the accuracy of estimated camera position in the global coordinate frame. It is defined as:

$$\mathbf{RTE} = \|\mathbf{t} - \mathbf{t}_{gt}\|_2. \tag{11}$$

This metric is expressed in meters and provides an interpretable measure of positional deviation. A lower RTE indicates a more accurate translation estimate, with an ideal value of zero corresponding to perfect alignment.

### A.1.2 RELATIVE ROTATION ERROR (RRE)

RRE evaluates the rotational misalignment between estimated rotation matrix $\mathbf{R}$ and ground-truth rotation matrix $\mathbf{R}_{gt}$. Specifically, it is computed based on the residual rotation that aligns the predicted orientation to the ground truth:

$$\mathbf{R}_{rel} = \mathbf{R}^{-1}\mathbf{R}_{gt}. \tag{12}$$

We convert the residual rotation matrix $\mathbf{R}_{rel}$ to Euler angles $\boldsymbol{\gamma} = [\gamma_1, \gamma_2, \gamma_3]^\top$ (typically using the ZYX convention), and define RRE as the sum of the absolute angular deviations along the three principal axes:

$$\mathbf{RRE} = \sum_{i=1}^{3} |\gamma(i)|. \tag{13}$$

RRE is reported in degrees, and a smaller value corresponds to higher rotational accuracy.

### A.1.3 SUMMARY

Together, RTE and RRE provide a comprehensive assessment of pose estimation accuracy. These metrics are particularly suitable for evaluating cross-modal registration methods, where both positional and orientational consistency between modalities are critical.

## A.2 CLASSICAL POSE SOLVER VS. OUR RAY-GUIDED POSE REGRESSION MODULE

To further analyze the stability of different pose estimation strategies, we provide visual comparisons between classical pose solver (Equation 3 and 4) and our ray-guided pose regression module. As shown in Figure 5, classical pose solver often produces unstable results when predicted rays are noisy or partially misaligned, leading to larger rotation errors and variance. This is because classical solvers rely heavily on fixed mathematical models and assumptions that do not adapt well to noisy or partially misaligned data. In contrast, our proposed module is learnable and thus can adapt to the specific characteristics of the input data. By learning from diverse examples, the module is able to handle noise and misalignment more effectively, producing more stable pose estimates even under challenging conditions. This adaptivity and robustness make our method more reliable, especially in real-world scenarios where noise, occlusion, and partial misalignment are common.

## A.3 ABLATION STUDIES ON PATCH SIZE

We conduct an ablation study to examine how the patch size associated with each ray influences registration performance, as shown in Table 5. Each ray corresponds to a $p \times p$ local image patch,

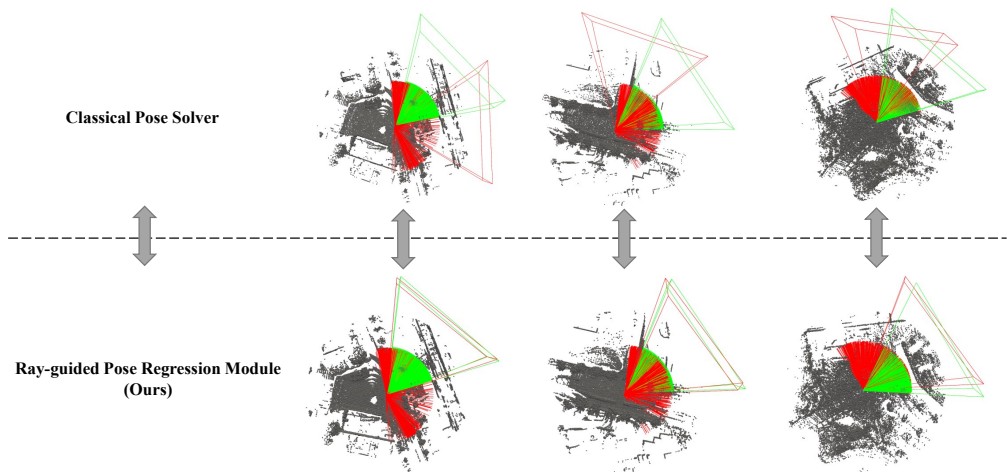

Figure 5: Visual comparison between classical pose solver and our proposed ray-guided pose regression module. Classical pose solver suffers from unstable predictions under noisy rays, whereas our method yields more robust and accurate pose estimations. Predicted rays and camera pose are visualized in red, while ground-truth rays and pose are shown in green.

controlling the spatial granularity of feature extraction. We find that a moderate patch size ($p = 8$) achieves the best performance, balancing spatial resolution and contextual coverage. Smaller patches ($p = 4$) lack context, while larger ones ($p = 16$ or $32$) reduce spatial precision, both leading to degraded performance.

Table 5: Ablation studies on patch size associated with each ray. Each ray corresponds to a $p \times p$ local image patch.

| $p$ | KITTI | | | nuScenes | | |
|---|---|---|---|---|---|---|
| | RTE(m)↓ | RRE(°)↓ | Acc(%)↑ | RTE(m)↓ | RRE(°)↓ | Acc(%)↑ |
| 4 | 0.12±0.09 | 0.85±0.96 | 99.34 | 0.42±0.31 | 1.84±6.06 | 94.71 |
| 8 | **0.09±0.08** | **0.63±0.71** | **99.75** | **0.39±0.29** | **1.48±5.72** | **96.61** |
| 16 | 0.21±0.15 | 0.95±0.93 | 99.43 | 0.43±0.30 | 1.82±6.02 | 95.25 |
| 32 | 0.18±0.14 | 0.83±0.89 | 99.34 | 0.44±0.31 | 2.44±6.04 | 90.31 |

## A.4 ABLATION ON FOCUS LOSS $\mathcal{L}_{foc}$

We investigate the impact of focus radius $\sigma$, which controls spatial constraints in cross-attention between patch and point features. As shown in Table 6, increasing $\sigma$ from 8 to 32 improves performance, suggesting that a broader local neighborhood provides richer 3D context for registration. However, further increasing $\sigma$ beyond 32 degrades performance, likely due to the inclusion of irrelevant or noisy points that weaken focused interactions. Overall, $\sigma = 32$ offers the best trade-off between geometric context and locality, and is adopted in all experiments.

## A.5 LIMITATIONS AND FUTURE WORK

While our method achieves competitive performance on challenging outdoor datasets, it still exhibits certain limitation primarily associated with the reliance on overlap prediction, which is a common challenge shared by almost all outdoor image-to-point cloud registration methods.

### A.5.1 ROBUST POSE ESTIMATION UNDER INACCURATE OVERLAP PREDICTION

Our approach incorporates a learned overlapping region detector module to guide cross attention between 2D image patches and 3D point clouds. In practice, we find that even when the predicted

Table 6: Ablation studies on focus radius $\sigma$. Here ✗ indicates method that doesn't use $\mathcal{L}_{foc}$.

| $\sigma$ | KITTI | | | nuScenes | | |
|---|---|---|---|---|---|---|
| | RTE(m)↓ | RRE(°)↓ | Acc(%)↑ | RTE(m)↓ | RRE(°)↓ | Acc(%)↑ |
| ✗ | 0.10±0.09 | 1.02±1.00 | 99.18 | 0.42±0.31 | 1.94±6.22 | 94.44 |
| 8 | 0.11±0.09 | 0.75±0.95 | 99.30 | 0.41±0.30 | 1.87±6.39 | 94.70 |
| 16 | 0.10±0.13 | 0.76±0.85 | 99.52 | 0.41±0.30 | 1.85±6.64 | 95.12 |
| 32 | 0.09±0.08 | **0.63±0.71** | **99.75** | **0.39±0.29** | **1.48±5.72** | **96.61** |
| 64 | **0.07±0.07** | 0.67±0.81 | 99.48 | 0.40±**0.29** | 1.79±6.60 | 95.81 |
| 128 | 0.09±0.09 | 0.91±0.86 | 99.43 | 0.41±**0.29** | 1.77±6.57 | 95.43 |
| 256 | 0.17±0.11 | 0.94±0.89 | 99.39 | 0.42±0.30 | 1.85±6.96 | 95.39 |

overlapping region is only partially correct, the model can still achieve accurate pose estimation as shown in Figure 6. This is primarily because cross attention mechanism is capable of selectively attending to informative rays within correctly predicted overlapping subset. Moreover, rays outside the predicted overlapping region can also benefit from their geometric and contextual relationships with rays inside the correctly predicted area, enabling accurate alignment despite partial supervision. These observations suggest that our method exhibits strong robustness to imperfect overlap prediction and does not rely heavily on highly accurate overlap masks to perform successful registration.

### A.5.2 FAILURE CASES UNDER COMPLETELY INCORRECT OVERLAP PREDICTION

In contrast to partially correct overlap predictions, when the predicted overlapping region is entirely incorrect—i.e., it contains no part of the true overlapping area—the model is fundamentally unable to establish any meaningful cross-modal interactions. Under this condition, the cross attention mechanism is misled and lacks access to any informative cues, resulting in failed ray-level reasoning across the modalities. Consequently, none of the predicted rays can be reliably estimated, and the final pose prediction becomes highly inaccurate, as shown in Fig. 7. This failure mode, although observed only in rare extreme cases, reveals a fundamental limitation of the current framework: when the predicted overlap region is entirely incorrect, the model lacks valid guidance for effective cross-modal interaction. Despite its infrequency, this issue highlights the dependency on overlap supervision and motivates future research toward overlap-independent pose estimation strategies.

### A.5.3 TOWARD OVERLAP-INDEPENDENT REGISTRATION

Although our framework exhibits a certain degree of robustness to overlap prediction errors, its performance still fundamentally depends on the overlap estimation being at least approximately correct. In future work, we plan to eliminate this reliance by designing overlap-free pose estimation methods. One promising future direction is to remove the explicit overlap prediction module altogether, allowing the network to implicitly learn to attend to the correct overlapping regions on its own. By leveraging latent cross-modal alignment cues, the model can infer relevant associations without relying on explicit supervision of overlap areas. Such an approach is expected to enhance both the robustness and generalization capability of the system, especially in challenging scenarios with ambiguous or noisy geometry.

### A.6 MORE IMPLEMENTATION DETAILS

Here we offer more implementation details about our proposed method. For feature extraction, we adopt backbones commonly used in existing approaches (Li et al., 2023). Specially, we adopt a 4-stage ResNet (He et al., 2016) as the image backbone, where the output channel dimension is 512. The output feature map is downsampled by a factor of 8 relative to the input image, yielding a resolution of $20 \times 64$ for KITTI and $20 \times 40$ for nuScenes. For the 3D backbone, we use a 4-stage KPConv (Thomas et al., 2019) where the output channel dimension is 512. The point clouds are voxelized with an initial voxel size of 15cm for both the KITTI dataset and nuScenes dataset. The batch size is set as 2. All experiments are conducted on a single RTX 3090 GPU. We implement our code using PyTorch 1.13.1 and CUDA 11.7.

**Overlap Prediction**          **Ray & Pose Prediction**

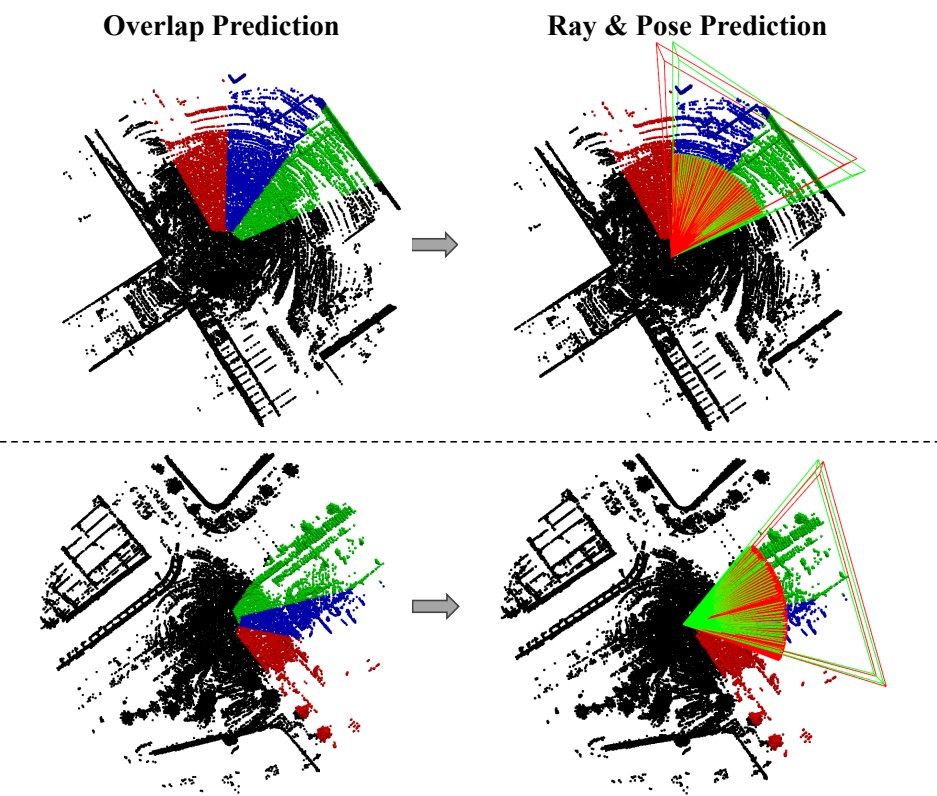

Figure 6: **Robust Pose Estimation Under Inaccurate Overlap Prediction.** Visualization of predicted overlapping regions and final registration results. Despite imperfect overlap prediction, our method achieves accurate pose estimation, showing strong robustness. Correctly identified overlapping points are shown in blue, missed overlaps in green, and false positives in red. Predicted rays and camera pose are visualized in red, while ground-truth rays and pose are shown in green.

### A.7 THE USE OF LARGE LANGUAGE MODELS (LLMS)

In this paper, we utilized ChatGPT, a large language model (LLM), to assist in the refinement and polishing of our writing. Specifically, ChatGPT was employed to improve the clarity, coherence, and overall presentation of the text. It helped with tasks such as rephrasing sentences, correcting grammatical errors, and suggesting improvements in academic tone. However, the contributions of the LLM were limited to writing assistance only and did not extend to the conceptualization, analysis, or development of the research. All ideas, methodologies, and experimental work presented in this paper were independently conceived and executed by the authors.

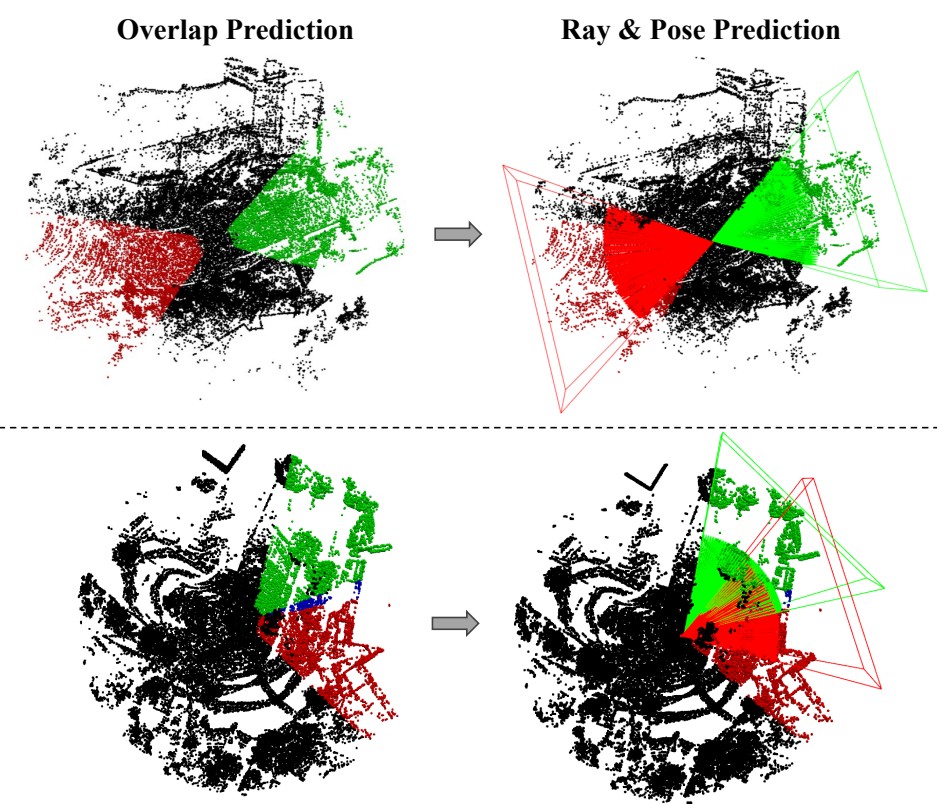

Figure 7: **Failure Cases Under Completely Incorrect Overlap Prediction**. Visualization of a rare but critical failure mode where the predicted overlapping region contains no part of the true overlapping area. In this extreme case, the cross attention module is misled and lacks any informative guidance due to completely incorrect overlap prediction, leading to failed ray reasoning and inaccurate pose estimation.

