# OpenReview forum: "RayI2P: Learning Rays for Image-to-Point Cloud Registration"
_ICLR.cc/2026/Conference — ICLR 2026 Poster_

### Official Review · Reviewer_JaMP · 2025-10-31

**Soundness:** 2
**Presentation:** 3
**Contribution:** 3
**Rating:** 4
**Confidence:** 4

**Summary:**

This paper addresses the problem of registration between a 3d point cloud and an image of the same scene. Compared to the problems of 3d registration between two
point clouds or image matching between two images, this problem is significantly more difficult. This difficulty comes from the ambiguities introduced by projecting 3d points
to images and finding the correct matches in the images. Learning based approaches have been dominating the field. The main novelty in this paper is the use of the rays to
parameterize the pose of the point cloud to the camera coordinate system. The rays are represented using Plücker coordinates, and this has been shown to be beneficial for
camera pose estimation and for 6d object pose estimation. This parameterization seems to be easier to learn and brings other benefits, like more precise alignment and
fewer projection ambiguities, and scale inconsistencies. The paper is well-written and not difficult to follow. The novelty seems to be limited, since all elements of the
proposed method have been known before and just combined using the ray representation. The overall pipeline has two stages composed of the ray prediction module, which basically does
3d point cloud and image feature fusion and is fairly standard and doesn't depend on the ray parameterization of the image patches. The second stage is a ray-guided regression module
that uses ray parameterization of the image patches, fused multi-modal features, and regresses the pose in terms of rotation and translation. The pipeline is supervised and applied to autonomous driving scenarios, where alignment between LIDAR scans and RGB images has been performed.

**Strengths:**

The paper is well-motivated and well-written. It was easy to follow it. Using ray parameterization for estimating the pose of the point cloud in the camera coordinate system
This makes sense, as it has been demonstrated for related problems, such as image pose estimation [1] and object pose estimation [2]. However, both approaches use this ray parameterization
for underlying diffusion models. Here, this is not the case. What I find interesting is the use of reference rays in the ray-guided pose regression. However, this is not well motivated.

**Weaknesses:**

The proposed method has limited novelty. The first part of the feature fusion, known as the ray prediction module, is not new and doesn't benefit from the camera ray parameterization; however, it utilizes fused features and predicts image patch rays. The motivation behind the use of the reference rays and predicted rays is not given. The paper
contains numerous ad-hoc design choices that are not clearly motivated or explained. See questions section for more details.
On the experimental side, I find the evaluation limited. It has been done only on two autonomous driving datasets. Why is this choice made for only this dataset and not some indoor datasets. Better motivation is needed for readers who are not interested in autonomous driving. I suppose the reason is that LIDAR and RGB images are not registered in the autonomous vehicles.

**Questions:**

Can you give more precise intuition behind using the reference rays? How do the guides and stabilize the regression process?

What will happen if you do not predict rays, but matches between 3d points and patches?


The results from Table 1 show that GraphI2P is actually performing very well and it is on par or better than the proposed method. What's the reason? What is the advantage of the proposed approach compared to GraphI2P?

In the ablation study, the results in Table 3 are a bit confusing. It is unclear what the fused features bring, as the rotation estimates in KITTI are more stable (smaller std) than in NuScenes. How do you explain this? What will happen if you have FPF and CPS only? RR seems to improve rotation estimation. Why?

---

> ### Author Response · Authors · 2025-11-18
> **Response Part 1**
>
> We sincerely thank the reviewer for the thorough evaluation, insightful comments, and recognition of our work’s strengths. Below we address each concern and question in detail, with additional clarifications on motivation, novelty, and experimental design.
> ## 1. Novelty of the proposed method
> ---
> We respectfully clarify that although some individual components (ray parameterization, feature fusion) have appeared in other contexts, our contribution lies in introducing a ray-based formulation tailored specifically to the unique challenges of image-to-point cloud registration, a setting fundamentally different from image-only pose estimation [1] or 6D object pose estimation [2].
>
> Prior ray-based works operate in intra-modal or dense-geometry settings where every predicted ray is expected to intersect valid scene surfaces. In contrast, image-to-point cloud registration involves cross-modal, sparse, incomplete, and partially-overlapping 3D observations. This creates an extremely challenging learning environment in which classical “camera-as-rays” pipelines [1] fail to converge. Our method introduces three key innovations to address these specific difficulties:
> 1. Patch-wise ray bundles tailored for sparse LiDAR, unlike pixel-level ray fields in image-only tasks.
> 2. Overlapping region detection + focus loss, which are necessary due to the extreme token imbalance (image patches: ~300; raw point cloud: up to 40,960 points). Without these mechanisms, the image tokens are overwhelmed by irrelevant 3D samples and training collapses.
> 3. A ray-guided pose regression module designed to aggregate noisy, incomplete ray–point evidence, a capability classical solvers fundamentally lack.
>
> Thus, while ray parameterization itself is not new, its integration into the cross-modal registration problem and the accompanying architectural changes are novel, necessary, and non-trivial.
>
> ## 2. Motivation and intuition behind reference rays
> ---
> The reviewer’s question highlights a critical design insight. We emphasize that **reference rays are the ground-truth (GT) of patch rays in the camera coordinate system**, and their core value lies in explicitly defining the target of pose transformation, while bringing additional geometric and learning benefits:
>
> ### Core intuition: Explicitly defining the alignment target
> - Predicted rays are estimated in the 3D point cloud (LiDAR) coordinate system, encoding how image patches project into the scene’s 3D geometry.
> - Reference rays are computed directly from known camera intrinsics (K) and patch coordinates (with R=I, t=0), representing the “true” projection direction and origin of each patch in the camera coordinate system (they are the GT ray representation for the camera’s local frame).
> - Our regression module’s core is to learn a 6-DoF transformation (R, t) that aligns predicted rays (LiDAR frame) to reference rays (camera frame). By embedding reference rays, we **directly provide the network with explicit “source-target” ray pairs** (predicted rays → reference rays) instead of forcing it to infer the alignment target from vague cross-modal cues alone.
>
> ### Key benefits beyond explicit alignment
> - **Eliminate ambiguous target inference**: Without reference rays, the network would have to guess both the transformation and the desired target ray directions (a two-fold ambiguity). Reference rays fix the target, allowing the model to focus solely on learning the transformation between coordinate frames thus reducing training complexity and convergence time.
> - **Strengthen geometric grounding**: Reference rays are derived from physical camera optics (intrinsics K), ensuring that the learned transformation adheres to real-world imaging geometry. This prevents the model from learning arbitrary feature mappings that perform well on training data but fail to generalize.
>
> ### Experimental validation of these benefits
> As shown in Table 1:
> -  KITTI RRE is 0.73° with a larger std (0.91), while adding reference rays reduces RRE to 0.63° (std=0.71), proving that explicit target alignment stabilizes rotation estimation.
> - On nuScenes (where ray noise and scene complexity are more pronounced), adding reference rays not only increases Acc by 2.36% and reduces RRE by 0.62°, but also drastically lowers RRE variance (from 8.44° to 5.72°). This confirms that reference rays’ geometric grounding not only improves accuracy but also enhances robustness and generalization to complex, noisy scenarios.
>
> ### Table 1: Comparison of without reference rays (RR) vs. our full method
>
> |        |         |    **KITTI**     |            |          |      **nuScenes**     |             |
> |:------:|:---------:|:----------:|:----------:|:----------:|:-----------:|:-----------:|
> |  | RTE (m)  | RRE (°)  | Acc (%) | RTE (m) | RRE (°) | Acc (%) |
> | w/o RR    | 0.10±0.09 | 0.73±0.91  | 99.41     | 0.41±0.29  | 2.10±8.44 | 94.25      |
> | **Ours** | **0.09±0.08** | **0.63±0.71** | **99.75** | **0.39±0.29** | **1.48±5.72** | **96.61** |

---

> > ### Author Response · Authors · 2025-11-18
> > **Response Part 2**
> >
> > ## 3. What happens if we directly predict 3D–2D matches?
> > ---
> > Predicting discrete 2D-3D matches (image patches ↔ 3D points) instead of rays is the core paradigm of classical matching-based methods, and CoFiI2P [3] is a representative baseline that adopts a LoFTR-inspired coarse-to-fine pipeline to establish patch-point correspondences, then estimates pose via EPnP-RANSAC, making it an ideal direct comparison for this question.
> >
> > As shown in Table 2, our ray-based approach outperforms CoFiI2P’s discrete matching paradigm across all metrics, highlighting the limitations of avoiding rays.
> >
> > In summary, abandoning ray prediction for discrete 2D-3D matches (as CoFiI2P does) reverts to inherent limitations of projection ambiguity and scale inconsistency, leading to higher errors, larger variance, and lower robustness. Our ray-based paradigm addresses these flaws, achieving superior performance.
> > ### Table 2: Comparison between Ray-Based (Ours) and Discrete Matching (CoFiI2P)
> >
> > |        |   **KITTI**     |            |     **nuScenes**     |             |
> > |:------:|:---------:|:----------:|:-----------:|:-----------:|
> > |  | RTE (m)  | RRE (°)  |  RTE (m) | RRE (°) |
> > | CoFiI2P    | 0.31±0.20 | 1.24±0.84  | 1.21±9.55  | 2.54±8.93 |
> > | **Ours** | **0.09±0.08** | **0.63±0.71** | **0.39±0.29** | **1.48±5.72** |
> >
> > ## 4. Why does GraphI2P perform comparable, and what is our advantage over it?
> > ---
> > ### Key reason for GraphI2P’s performance
> > GraphI2P [4] relies on a pre-trained external depth estimation model [5] (trained on large-scale datasets beyond the target registration benchmarks) to generate dense depth maps from RGB images. This allows it to reconstruct a dense image-derived point cloud by combining depth maps with camera intrinsics. Instead of aligning RGB images directly to LiDAR point clouds (cross-modal), it only needs to match two point clouds (image-derived + LiDAR). Additionally, it designs graph-based matching strategies tailored for point cloud correspondence learning, which further boosts its performance on this converted task.
> >
> > ### Our core advantages over GraphI2P
> > - No external dependencies, better generalizability: GraphI2P’s performance is heavily tied to the quality of the external depth estimator. It fails to generalize to scenes with unseen depth distributions where the pre-trained depth model degrades. Our method uses only RGB images and LiDAR point clouds, eliminating reliance on external tools and adapting more flexibly to diverse scenarios.
> > - Higher efficiency: The external depth estimation step adds significant computational overhead (e.g., inference time for depth models like DPT [6] or MiDaS [7]). Our end-to-end ray-based framework integrates cross-modal alignment and pose regression into a single pipeline, achieving much faster inference without compromising accuracy.
> > - Pure cross-modal alignment capability: Unlike GraphI2P, which avoids the RGB-LiDAR modality gap entirely, our method directly models and resolves it via ray-based representation. Rays encode the intrinsic geometric relationship between 2D image patches and 3D LiDAR points, providing finer-grained supervision than GraphI2P’s image-derived point clouds (which are indirect and distorted by depth estimation errors). This enables more precise alignment of appearance (RGB) and geometry (LiDAR) cues, leading to better performance on pure cross-modal tasks where depth conversion is not an option.

---

> > > ### Author Response · Authors · 2025-11-18
> > > **Response Part 3**
> > >
> > > ## 5. Clarifications on the ablation study
> > > ---
> > > ### Confused about FPF-only results in ablation study
> > > As shown in Table 3, FPF-only performs nearly identically to the overlapping region detector (ORD) on both datasets. It confirms that fused patch features (FPF) alone cannot provide sufficient geometric information for reliable pose regression. They only refine the results of overlapping region detector (ORD) slightly, but fail to deliver the critical geometric cues needed to reduce ambiguity in rotation estimation.
> > >
> > > The confusion about the larger RRE variance on nuScenes (10.05° for FPF-only vs. 1.67° on KITTI) stems directly from dataset-specific ORD performance: nuScenes features denser urban environments, dynamic objects, and extreme depth variations, making overlapping region detection far more challenging. ORD itself has a much larger RRE variance on nuScenes (10.51°) than KITTI (1.72°), and since FPF-only relies entirely on ORD’s output (without rays to provide additional geometric constraints), this variance propagates directly to the final rotation results. In short, the larger variance is not a flaw of FPF, but a reflection of ORD’s difficulty in learning reliable overlapping regions on complex scenes, something FPF alone cannot compensate for.
> > >
> > > ### Table 3: Performance Comparison of ORD vs. FPF-only (No Predicted/Reference Rays)
> > >
> > > |        |         |    **KITTI**     |            |          |      **nuScenes**     |             |
> > > |:------:|:---------:|:----------:|:----------:|:----------:|:-----------:|:-----------:|
> > > |  | RTE (m)  | RRE (°)  | Acc (%) | RTE (m) | RRE (°) | Acc (%) |
> > > |  ORD    | 0.46±0.31 | 2.05±1.72  | 94.13     | 0.88±1.19  | 4.71±10.51 | 75.39      |
> > > | FPF-only | **0.33±0.18** | **2.07±1.67** | **94.48** | **0.48±0.32** | **3.23±10.05** | **89.79** |
> > >
> > > ### What if we have FPF and CPS only?
> > >
> > > Classical pose solvers (CPS, details in the main text) require only predicted rays and camera intrinsics rather than high-dimensional FPF to obtain camera pose. FPF lacks explicit geometric interpretability, making "FPF + CPS" fundamentally unworkable.
> > >
> > > We wonder if we misunderstand the reviewer’s intended setup for this combination.
> > > Please let us know if you had a specific configuration in mind, and we can supplement corresponding experiments.
> > >
> > > ### Why RR improves rotation estimation
> > >
> > > Unlike FPF, which is tied to overlapping region detection (ORD) , reference rays (RR) are derived from camera intrinsics (physical optics) and provide a stable directional prior.
> > > It converts vague cross-modal feature similarity into explicit “source-target” ray alignment (predicted rays → reference rays), creating a consistent supervision signal that reduces rotation errors.
> > >
> > > ## 6. On the choice of autonomous driving datasets
> > > ---
> > > Our choice of KITTI and nuScenes is based on three practical reasons:
> > > 1. Task relevance: Autonomous driving is the primary real-world application for image-LiDAR registration, where the modality gap (dense RGB vs. sparse LiDAR) and environmental complexity (dynamic objects, varying depth, occlusions) are most pronounced. These datasets provide large-scale, diverse data that rigorously test generalization.
> > > 2. Benchmark standardization: KITTI and nuScenes are the de facto benchmarks for image-to-point cloud registration, with well-defined evaluation protocols (RTE/RRE/Acc) and public baselines (GraphI2P, ICL, VP2P-match [8]) that enable fair comparison. Indoor datasets typically have denser point clouds, smaller depth variations, and fewer occlusions, reducing the modality gap that our method is designed to address.
> > > 3. Practical impact: Most registration methods for indoor scenes focus on RGB-D (not RGB-LiDAR) alignment, where depth maps reduce ambiguity. Our work targets the harder RGB-LiDAR setting, which is critical for autonomous driving (LiDAR is standard, RGB-D is not).
> > >
> > >
> > >
> > > We thank the reviewer again for the thorough evaluation and insightful questions. We have clarified the novelty and motivation of our ray-based formulation, explained the necessity of reference rays and the regression module, addressed comparisons to GraphI2P, clarified ablation behaviors, and justified the experimental setup. We hope these detailed explanations address the reviewer’s concerns and demonstrate the practical significance and methodological soundness of our work.

---

> > > > ### Author Response · Authors · 2025-11-18
> > > > **Reference**
> > > >
> > > > [1] Jason Y. Zhang, Amy Lin, Moneish Kumar, Tzu-Hsuan Yang, Deva Ramanan, and Shubham Tulsiani. Cameras as rays: Pose estimation via ray diffusion. In The Twelfth International Conference on Learning Representations, 2024.
> > > >
> > > > [2] Huang, Junwen, Shishir Reddy Vutukur, Peter KT Yu, Nassir Navab, Slobodan Ilic, and Benjamin Busam. "RayPose: Ray Bundling Diffusion for Template Views in Unseen 6D Object Pose Estimation." In *Proceedings of the IEEE/CVF International Conference on Computer Vision*, pp. 9102-9112. 2025.
> > > >
> > > > [3] Shuhao Kang, Youqi Liao, Jianping Li, Fuxun Liang, Yuhao Li, Xianghong Zou, Fangning Li, Xieyuanli Chen, Zhen Dong, and Bisheng Yang. Cofii2p: Coarse-to-fine correspondences-based image to point cloud registration. IEEE Robotics and Automation Letters, 2024.
> > > >
> > > > [4] Lin Bie, Shouan Pan, Siqi Li, Yining Zhao, and Yue Gao. Graphi2p: Image-to-point cloud registration with exploring pattern of correspondence via graph learning. In Proceedings of the Computer Vision and Pattern Recognition Conference, pp. 22161–22171, 2025.
> > > >
> > > > [5] Shariq Farooq Bhat, Reiner Birkl, Diana Wofk, Peter Wonka, and Matthias M¨uller. Zoedepth: Zero-shot transfer by combining relative and metric depth. arXiv preprint arXiv:2302.12288, 2023.
> > > >
> > > > [6] Ren´e Ranftl, Alexey Bochkovskiy, and Vladlen Koltun. Vision transformers for dense prediction. In Proceedings of the IEEE/CVF International Conference on Computer Vision (ICCV), pages 12179–12188, 2021.
> > > >
> > > > [7] Ren´e Ranftl, Katrin Lasinger, David Hafner, Konrad Schindler, and Vladlen Koltun. Towards robust monocular depth estimation: Mixing datasets for zero-shot cross-dataset transfer. IEEE Transactions on Pattern Analysis and Machine Intelligence (TPAMI), 2020.
> > > >
> > > > [8] Zhou, Junsheng, Baorui Ma, Wenyuan Zhang, Yi Fang, Yu-Shen Liu, and Zhizhong Han. "Differentiable registration of images and lidar point clouds with voxelpoint-to-pixel matching." *Advances in Neural Information Processing Systems* 36 (2023): 51166-51177.

---

### Official Review · Reviewer_RNei · 2025-10-31

**Soundness:** 3
**Presentation:** 3
**Contribution:** 3
**Rating:** 6
**Confidence:** 2

**Summary:**

This paper proposes an image-to-point cloud registration method that estimates the 6-DoF camera pose of a query image relative to a 3D point cloud map. The proposed ray-based registration framework first predicts patch-wise 3D ray bundles connecting image patches to the 3D scene, then estimates the camera pose via a differentiable ray-guided regression module.

**Strengths:**

- The proposed ray-based registration framework for image-to-point cloud registration is interesting.
- The experimental results verified the effectiveness of the proposed method.

**Weaknesses:**

- Additional discussion comparing the proposed method with other ray-based representation methods [1] should be added.
- The approach appears to be a direct application of ray-based representation for pose estimation to the image-to-point cloud registration task. The specific challenge this addresses should be further clarified.

[1] Jason Y. Zhang, Amy Lin, Moneish Kumar, Tzu-Hsuan Yang, Deva Ramanan, and Shubham Tulsiani. Cameras as rays: Pose estimation via ray diffusion. In *The Twelfth International Conference on Learning Representations*, 2024.

**Questions:**

What is the key challenge in image-to-point cloud registration compared to ray-based representation for pose estimation?

---

> ### Author Response · Authors · 2025-11-18
> **Response Part 1**
>
> We thank the reviewer for the constructive feedback and for recognizing the novelty and effectiveness of our ray-based image-to-point cloud registration framework. Below we address the raised concerns and provide additional clarification and analysis.
>
> ## 1. Comparison with other ray-based representation methods [1]
> ---
> We appreciate the reviewer’s suggestion to clarify the connection with existing ray-based works. A critical distinction to emphasize is that our method is the first to introduce ray-based representation into the image-to-point cloud registration task. All prior ray-based works (including [1]) are designed for entirely different task settings and solve distinct problems, making direct performance comparison infeasible.
>
> Since no existing ray-based method is designed for or evaluated on image-to-point cloud registration benchmarks (e.g., KITTI, nuScenes), there is no overlapping experimental setup or metric to support direct performance comparison. Our key contribution here is not “improving existing ray-based methods” but “pioneering the application of ray-based representation to solve the long-standing cross-modal challenges in image-to-point cloud registration”, a novel direction that fills the gap between ray-based geometric modeling and cross-modal alignment tasks.
>
> ## 2. Clarification on specific challenges addressed by the proposed approach
> ---
> The reviewer raises an important point regarding whether our method goes beyond directly applying ray-based pose estimation to image-to-point cloud registration. In practice, such a direct application is fundamentally infeasible due to the severe modality imbalance and noise characteristics of LiDAR point clouds.
>
> Ray-based pose estimation methods operate purely on images, where each input contains a few hundred structured tokens with dense spatial coherence. In contrast, our task introduces an additional modality containing up to 40,960 raw 3D points, most of which are irrelevant to the visible image region and many of which arise from occluded, distant, or non-overlapping areas. When these two modalities are fused naïvely, the massive discrepancy in token count causes the attention mechanism to collapse, and we observe that training fails to converge because image patches are overwhelmed by noisy and semantically unrelated 3D samples. This is why we introduce two key components (overlapping region detection and the focus loss) to explicitly filter irrelevant points and guide image patches to attend to geometrically meaningful regions. Even after filtering, the point cloud still contributes several thousand tokens, reinforcing the need for these targeted mechanisms.
>
> Moreover, classical geometric solvers used in existing ray-based pose estimation are highly sensitive to noise and degeneracies. To address this limitation, we design a ray-guided pose regression module that jointly integrates patch rays and reference rays and learns to suppress inconsistent geometric cues, producing noticeably more stable predictions than classical solvers (with visual examples and detailed analysis in Appendix A.2).

---

> > ### Author Response · Authors · 2025-11-18
> > **Response Part 2**
> >
> > ## 3. Key challenge in image-to-point cloud registration compared to ray-based pose estimation
> > ---
> > The key challenge in image-to-point cloud registration, compared with ray-based pose estimation, comes from the fact that the two tasks rely on completely different types of geometric information. In ray-based pose estimation, every predicted ray is expected to correspond to some valid surface in the scene because the model operates only on images, and dense pixel information allows the network to infer consistent geometry everywhere in the view. This makes the underlying geometry effectively dense and well-behaved.
> >
> > In image-to-point cloud registration, this assumption breaks down entirely. A LiDAR point cloud is extremely sparse and incomplete: large parts of the scene contain no points at all, and many 3D points come from areas that are not visible in the query image. As a result, when we predict a ray from an image patch, there is a high chance that this ray does not hit any corresponding 3D point, not because the prediction is wrong, but simply because the LiDAR did not sample that region. This creates many ambiguous or degenerate ray–point relationships. Moreover, because the point cloud contains tens of thousands of unordered points, most of which are irrelevant to the image, the model must identify the small subset that actually overlaps with the camera’s view and ignore the rest. None of these difficulties occur in image-only ray-based pose estimation, where dense appearance and geometry ensure that most rays provide meaningful constraints. Therefore, the primary challenge is that in image-to-point cloud registration the model must extract reliable geometric signals from a very sparse, noisy, and partially overlapping 3D point set.
> >
> > We thank the reviewer again for pushing us to clarify the practical problem-solving value of our work. We have explicitly defined concrete challenges in image-to-point cloud registration, detailed how our ray-based framework resolves each. We are happy to provide any further clarification if needed.
> >
> > [1] Jason Y. Zhang, Amy Lin, Moneish Kumar, Tzu-Hsuan Yang, Deva Ramanan, and Shubham Tulsiani. Cameras as rays: Pose estimation via ray diffusion. In The Twelfth International Conference on Learning Representations, 2024.

---

### Official Review · Reviewer_6CqR · 2025-11-01

**Soundness:** 3
**Presentation:** 3
**Contribution:** 3
**Rating:** 8
**Confidence:** 3

**Summary:**

This paper proposed a method for image-to-point cloud registration. Matching-based approaches is challenged by projection ambiguity and depth-induced scale inconsistency. To adress these problems, they introduce a differentiable ray-guided regression module which regress camera pose from predicted Plu ̈cker rays, thereby naturally resolving projection ambiguity and depth-induced scale ambiguity. The authors conduct experiments on KITTI and nuScenes. Method are evaluated by Relative Translation Error (RTE), average Relative Rotation Error (RRE), and registration accuracy (Acc). Compared to existing state-of-the-art approaches, the proposed method achieves strong accuracy, while remaining computationally efficient.

**Strengths:**

This paper is clearly motivated and well written. The paper intoduces a novel ray-guided pose regression module which addresses projection ambiguity and depth-induced scale inconsistency. Its technical rationale is supported by findings in the generalized camera models. The experimental setup is comprehensive, covering multiple datasets, metrics and baselines.

**Weaknesses:**

- The author did not analyze the causes of errors in the predicted rays. The neural regression module seems more like a compensatory measure; however, its use may raise concerns about generalization.
- Some of the illustrative figures are not very clear. In Figure 3, does the color of the points represent the depth error or the actual depth value?

**Questions:**

- Can the ray errors could be mitigated using diffusion-based methods?
- If the error threshold for Acc is set smaller, can the regression-based method achieve precision comparable to that of matching-based methods?

---

> ### Author Response · Authors · 2025-11-18
> **Response Part 1**
>
> We thank the reviewer for the constructive comments and positive evaluation. Below we address each point with clarified explanations and additional results.
>
> ## 1. Causes of errors in predicted rays
> ---
> The analysis of ray prediction failures is included in Appendix A.5.2 of the original submission. The primary cause of inaccurate ray predictions is erroneous *overlapping region detection*. Outdoor point clouds contain a large number of irrelevant points that do not overlap with the query image. Therefore, most image-to-point cloud registration pipelines, including ours, begin by predicting which points lie inside the overlapping area.
>
> In challenging cases (frequently observed in nuScenes), the predicted overlap region may significantly deviate from the ground truth. In extreme failure modes, the predicted and actual overlap sets share little or no intersection, leaving the ray prediction module with no meaningful 3D points to interact with, which directly leads to inaccurate rays.
>
> Appendix A.5.2 of the original submission provides detailed analyses and failure visualizations of these failure cases. We also report the pose accuracy produced by the overlap-region detector (ORD) versus our full method in Table 1. The substantial gap demonstrates that our framework can correct moderate overlap-region prediction errors (with detailed analyses and visualizations in Appendix A.5.1), but large overlap failures inevitably propagate to the ray-prediction stage.
>
> ### Table 1: Comparison of ORD vs. our full method
>
> |        |         |    **KITTI**     |            |          |      **nuScenes**     |             |
> |:------:|:---------:|:----------:|:----------:|:----------:|:-----------:|:-----------:|
> |  | RTE (m)  | RRE (°)  | Acc (%) | RTE (m) | RRE (°) | Acc (%) |
> | ORD    | 0.46±0.31 | 2.05±1.72  | 94.13     | 0.88±1.19  | 4.71±10.51 | 75.39      |
> | **Ours** | **0.09±0.08** | **0.63±0.71** | **99.75** | **0.39±0.29** | **1.48±5.72** | **96.61** |
>
> ## 2. Purpose and generalization of the regression module
> ---
> The regression module is not a compensatory component but a robust geometric aggregator designed to stabilize pose estimation when rays contain moderate noise (an unavoidable condition in outdoor I2P tasks). Analytical solvers are highly sensitive to deviations from ideal ray geometry; when overlap-region detection introduces small inconsistencies, classical solvers may become unstable or produce high-variance results.
>
> In contrast, the regression module jointly integrates patch rays and reference rays and learns to suppress inconsistent geometric cues.
>
> Appendix A.2 of the original submission provides visual examples and detailed analysis, illustrating cases where the regression module produces noticeably more stable predictions than classical solvers. Quantitatively, Table 2 shows that the regression module consistently lowers the mean rotation error and substantially reduces variance (especially on nuScenes where ray noise is more pronounced), demonstrating improved robustness and generalization.
>
> ### Table 2: Classical Pose Solver (CPS) vs. our Regression Module
>
> |        |         |    **KITTI**     |            |          |      **nuScenes**     |             |
> |:------:|:---------:|:----------:|:----------:|:----------:|:-----------:|:-----------:|
> |  | RTE (m)  | RRE (°)  | Acc (%) | RTE (m) | RRE (°) | Acc (%) |
> | CPS    | 0.10±**0.08** | 0.82±0.78 | 99.62 | **0.39±0.28** | 2.51±20.93 | 94.49 |
> | **Ours** | **0.09±0.08** | **0.63±0.71** | **99.75** | **0.39**±0.29 | **1.48±5.72** | **96.61** |
>
> ## 3. Clarification regarding Figure 3
> ---
> The caption has been updated in the revised manuscript to explicitly state that point colors represent the *actual depth value*.
>
> ## 4. Diffusion-based refinement of ray errors
> ---
> Since most ray failures arise from erroneous overlap-region detection rather than from uncertainty in ray prediction itself, diffusion-based refinement of rays would not address the primary failure mode. Moreover, diffusion introduces significant computational overhead, whereas our method is designed to provide strong accuracy while remaining efficient. We thus believe diffusion is not advantageous for this task.

---

> > ### Author Response · Authors · 2025-11-18
> > **Response Part 2**
> >
> > ## 5. Accuracy under smaller thresholds and comparison with matching-based methods
> > ---
> > To address the reviewer’s question, we further compared our method against the most recent open-sourced matching-based approach, ICL [1], under stricter translation and rotation thresholds.
> >
> > Results in Table 3 and Table 4 show that our approach maintains consistently higher accuracy across all tested thresholds, including challenging fine-grained settings. This confirms that our regression-based formulation achieves superior precision relative to matching-based pipelines.
> >
> > ### Table 3: Acc (%) under Different Thresholds (Ours / ICL) on KITTI dataset
> >
> > | | t = 2.0m | t = 1.5m | t = 1.0m | t = 0.5m |
> > |:-----------------:|:--------:|:--------:|:--------:|:--------:|
> > | **r = 5°** | **99.75** / 97.49 | **99.75** / 97.47 | **99.71** / 97.40 | **99.48** / 94.11 |
> > | **r = 4°** | **99.61** / 95.58 | **99.61** / 95.58 | **99.57** / 95.51 | **99.34** / 92.87 |
> > | **r = 3°** | **99.12** / 91.73 | **99.12** / 91.73 | **99.09** / 91.67 | **98.85** / 89.67 |
> > | **r = 2°** | **96.83** / 84.15 | **96.83** / 84.15 | **96.81** / 84.12 | **96.63** / 82.79|
> > | **r = 1°** | **80.86** / 62.55 | **80.86** / 62.55 | **80.86** / 62.55 | **80.78** / 61.89 |
> >
> > ### Table 4: Acc (%) under Different Thresholds (Ours / ICL) on nuScenes dataset
> >
> > | | t = 2.0m | t = 1.5m | t = 1.0m | t = 0.5m |
> > |:-----------------:|:--------:|:--------:|:--------:|:--------:|
> > | **r = 5°** | **96.61** / 90.94 | **96.17** / 89.17 | **93.91** / 81.86 | **72.75** / 50.69 |
> > | **r = 4°** | **95.21** / 86.40 | **94.78** / 85.12 | **92.56** / 78.24 | **72.00** / 48.74 |
> > | **r = 3°** | **92.04** / 79.67 | **91.63** / 78.48 | **89.48** / 72.25 | **70.16** / 45.22 |
> > | **r = 2°** | **84.45** / 66.42 | **84.08** / 65.45 | **82.21** / 60.46 | **65.18** / 38.56 |
> > | **r = 1°** | **61.80** / 39.73 | **61.56** / 39.11 | **60.21** / 36.35 | **48.52** / 23.55 |
> >
> > We thank the reviewer again for the thoughtful comments. We have answered all raised concerns with detailed explanations and supporting evidence, and we hope the responses above fully resolve the questions posed. We remain available to provide any further clarification if needed.
> >
> > [1] Xinjun Li, Wenfei Yang, Jiacheng Deng, Zhixin Cheng, Xu Zhou, and Tianzhu Zhang. Implicit correspondence learning for image-to-point cloud registration. In Proceedings of the IEEE/CVF Conference on Computer Vision and Pattern Recognition, pp. 16922–16931, 2025.

---

### Author Response · Authors · 2025-11-30
**Consolidated Rebuttal: Key Points & Clarifications**

To assist readers in efficiently understanding how our rebuttal addresses the reviewers’ concerns, we provide this short summary that consolidates the main points of our responses.

### **Background for the Area Chair:**
- As per the ICLR organizing committee's instructions, since the official reviewer discussion has been halted, we provide this summary to ensure all our detailed responses and new evidence are readily accessible for your meta-review.
- The initial reviews were broadly positive: reviewers highlighted the clarity of motivation, the soundness of the proposed ray-based formulation, the strong empirical results on two established benchmarks, and the overall quality of the presentation. The initial evaluation scores reflected this generally favorable assessment.
- Their main questions focused on clarifying specific design choices, the positioning of novelty relative to prior ray-based works, and understanding the behavior of several ablation components, all of which we addressed thoroughly in our responses.


During the rebuttal period, we prepared comprehensive and detailed answers to all reviewer comments, including additional analyses, new experiments, extended ablations, and clarifications of the methodological intuition and novelty.
As the discussion could not continue beyond our submitted responses, we aimed to ensure that each concern was addressed as thoroughly and self-contained as possible, so that the evaluation can proceed smoothly based on the information provided.

Below we summarize the key resolutions for clarity:
- **Technical & Ablation Clarifications:** We analyzed the sources of ray-prediction errors , explained the motivation and geometric intuition behind the reference rays (showing they stabilize pose estimation by providing an explicit alignment target ), and described how the regression module is a robust geometric aggregator (not a compensatory measure) that stabilizes pose estimation against noise.
- **New Experiments & Evidence:** We conducted new comparisons against matching-based methods (ICL) under stricter thresholds, confirming that our regression-based formulation achieves superior precision. We also provided additional ablations, and clarified the reasons for choosing challenging autonomous driving datasets (KITTI/nuScenes).
- **Clarification of Novelty & Challenges:** We clarified how prior “camera-as-rays” methods differ fundamentally from the cross-modal I2P setting, and why existing ray-based pose estimation pipelines cannot be directly applied. Also, we highlighted the specific challenge (sparsity, noisy, partial overlap, and modality imbalance) that our design resolves.
- **Comparisons with Related Methods:** We provided detailed explanations and new experiments comparing our framework with depth-dependent GraphI2P and other matching-based pipelines, demonstrating the benefits of our ray-based formulation in accuracy, robustness, and architectural simplicity.

Our intention with this summary is to make the structure of our responses easier to follow, especially since no reviewer discussion occurred during the response period. We hope this consolidated overview helps all readers form a clear understanding of how each concern has been addressed.

We sincerely appreciate the time and effort invested in evaluating our submission, and we hope that the clarity and completeness of our responses support a smooth and informed assessment process.

---

### Meta-Review · Area_Chair_mwRp · 2025-12-15

**Summary:**

This paper receives 1x marginally above the acceptance threshold, 2x accept, good paper, and 1x marginally below the acceptance threshold.

Reviewers 6CqR and RNei, who gave an accept rating, praised the novel ray-based registration approach for image-to-point cloud alignment, noting that it effectively addresses projection ambiguity and depth-induced scale inconsistency. The ray-guided regression module is seen as a robust solution, and the paper achieves state-of-the-art performance on KITTI and nuScenes. The experimental results are strong, with detailed ablation studies supporting the design choices.

Reviewer JaMP, who rated the paper marginally below the threshold, raised concerns about the incremental novelty of the approach, as many components are not entirely new. They also questioned the motivation for using reference rays and the limited generalization to other datasets beyond autonomous driving. Additionally, the comparison to other ray-based methods was seen as lacking. In response, the authors clarified that their method is the first to apply ray-based representation to image-to-point cloud registration, addressing the specific challenges of sparse 3D data and image modality gaps. They also provided further clarification on the role of reference rays and justified their dataset choice based on the relevance to autonomous driving scenarios.

The metareviewer considers the reviewers' concerns but concludes that the authors have addressed them adequately, especially with the clarification of novelty and design choices. Given the paper's strong experimental results and the substantial contribution to the field, the metareviewer recommends acceptance.

**Reviewer Concerns:**

Concerns addressed by the rebuttal:

1. The authors clarified that their method is the first to apply ray-based representation specifically to image-to-point cloud registration, filling a gap in addressing sparse 3D data and the image modality gap. This clarification helped address concerns about the incremental novelty of the approach.

2. The authors explained that reference rays serve as geometric anchors to stabilize pose estimation, which addressed concerns about the lack of motivation for their use.

3. The authors justified using KITTI and nuScenes datasets, emphasizing their relevance to autonomous driving, where LiDAR and RGB data are often misaligned. This addressed concerns about the generalization of the method to other domains and datasets.

4. The authors clarified that their work differs significantly from previous ray-based pose estimation methods, as it targets a specific task (image-to-point cloud registration) and resolves challenges like sparse and noisy data.

Concerns still outstanding:

Limited Novelty of Some Components: While the authors clarified the overall novelty of the application, some reviewers felt that components like the feature fusion and ray-guided regression module were not entirely new in isolation, and the methodology’s incremental nature still remains a concern for some.

**Reviewer Scores:**

Reviewer 6CqR and RNei may maintain their score. Reviewer JaMP may have raised their score following the rebuttal, but lingering concerns about limited novelty and experimental scope could keep their final rating near the acceptance threshold.

---

### Decision · Program_Chairs · 2026-01-26

Accept (Poster)